# Predictors and Profile of Severe Infectious Complications in Multiple Myeloma Patients Treated with Daratumumab-Based Regimens: A Machine Learning Model for Pneumonia Risk

**DOI:** 10.3390/cancers16213709

**Published:** 2024-11-03

**Authors:** Damian Mikulski, Marcin Kamil Kędzior, Grzegorz Mirocha, Katarzyna Jerzmanowska-Piechota, Żaneta Witas, Łukasz Woźniak, Magdalena Pawlak, Kacper Kościelny, Michał Kośny, Paweł Robak, Aleksandra Gołos, Tadeusz Robak, Wojciech Fendler, Joanna Góra-Tybor

**Affiliations:** 1Department of Biostatistics and Translational Medicine, Medical University of Lodz, 92-215 Lodz, Poland; damian.mikulski@umed.lodz.pl (D.M.); kacper.koscielny@umed.lodz.pl (K.K.); wojciech.fendler@umed.lodz.pl (W.F.); 2Department of Hematooncology, Provincial Multi-Specialized Oncology and Trauma Center, 93-513 Lodz, Poland; marcin.kedzior@stud.umed.lodz.pl (M.K.K.); katarzyna.jerzmanowska1@stud.umed.lodz.pl (K.J.-P.); michal.kosny@stud.umed.lodz.pl (M.K.); pawel.robak@umed.lodz.pl (P.R.); aleksandra.golos@umed.lodz.pl (A.G.); 3Department of Hematology, Medical University of Lodz, 90-419 Lodz, Poland; zaneta.witas@stud.umed.lodz.pl (Ż.W.); lukasz.wozniak3@stud.umed.lodz.pl (Ł.W.); magdalena.pawlak@stud.umed.lodz.pl (M.P.); tadeusz.robak@umed.lodz.pl (T.R.); 4Department of General Hematology, Copernicus Memorial Hospital, 93-513 Lodz, Poland

**Keywords:** adverse event, daratumumab, infection, multiple myeloma, PDW, platelet distribution width

## Abstract

Our research explores the profile and risk factors for infections in multiple myeloma patients undergoing treatment with daratumumab, a key drug in chemotherapy regimens for this disease. The study seeks to identify which patients are at the highest risk of developing severe infections and the factors contributing to this risk, as infections are a major concern for these patients. Analysis of patient data from our facility showed that lower hemoglobin levels and poorer performance status significantly increase the risk of serious infections. Additionally, we developed predictive algorithms to identify individuals at elevated risk of developing pneumonia during treatment. The findings from our study may help healthcare providers identify high-risk patients and implement targeted strategies to prevent infections, ultimately improving patient care.

## 1. Introduction

Multiple myeloma (MM) is a hematologic malignancy characterized by the clonal proliferation of plasma cells in the bone marrow and in extramedullary organs, with the secretion of monoclonal (M) protein [1]. MM is the second most common hematological malignancy globally that tends to occur with an estimated annual incidence of 7.2 per 100,000 men and women [2].

Daratumumab (Dara) is a CD38-targeting human immunoglobulin G kappa (IgG1κ) monoclonal antibody with a well-characterized mechanism of action. Briefly, it binds with high affinity to a specific CD38 epitope on CD38-expressing MM cells and induces complement-dependent cytotoxicity (CDC), antibody-dependent cell-mediated cytotoxicity (ADCC), and antibody-dependent cellular phagocytosis (ADCP) against myeloma cells [3]. In clinical trials regarding both newly diagnosed MM (NDMM) and relapsed/refractory MM (RRMM) patients, adding Dara to the protocol boosted the depth of response, including negativity of minimal residual disease (MRD), which led to longer median progression-free survival (PFS) [4,5,6,7,8]. Currently, Dara is the backbone of the recommended therapies for both NDMM and RRMM [9].

Infections remain a significant cause of death in patients with MM. In studies conducted during the previous years when chemotherapy was the main form of treatment, infections contributed to premature death in up to 14% to 45% of patients [10,11]. Blimark et al. described a sevenfold increase in the risk of bacterial infections and a tenfold increase in the risk of viral infections in patients with MM [12]. Despite improved outcomes, adding Dara into treatment regimens also resulted in a higher incidence of infectious complications (ICs). In a recent meta-analysis of 11 randomized controlled trials that included patients treated with anti-CD38 monoclonal antibodies, the cumulative incidence of any grade infections and severe infections were over 75% and 25%, respectively [13]. Especially patients treated with ani-CD38 antibodies had almost a 40% higher risk of pneumonia and severe pneumonia [13]. It is postulated that CD38 influences a pro-inflammatory response in immune cells, impacting leukocyte recruitment, macrophage function, and dendritic cell migration, which impairs T and B cell activity [14]. This and the depletion of NK cells and other CD38-expressing immune cells by daratumumab may explain the immunosuppression observed in some patients [15]. However, identifying patients at risk of ICs for individualized prophylaxis remains challenging. Recently, van de Donk et al., using data from the ALCYONE and MAIA trials, developed a predictive model to identify serious infection risks in non-transplant eligible NDMM patients, focusing on factors like age over 75, high lactate dehydrogenase (LDH), low serum albumin, and elevated alanine transaminase (ALT) [16]. It should be noted that these risk factors might not apply to NDMM transplant-eligible and RRMM patients, where high tumor burden and other unfavorable factors are more relevant.

Our study aimed to assess the frequency and risk factors of ICs in patients treated with Dara in a real-world setting, including those with ND and RRMM across various Dara-based regimens. Additionally, we sought to develop a straightforward, universal predictive model to identify high-risk patients for ICs, particularly pneumonia, facilitating easy implementation in clinical settings for preemptive interventions.

## 2. Materials and Methods

### 2.1. Patients and Treatment

This retrospective, real-life study included MM patients treated with Dara-based regimens according to the Polish Ministry of Health’s drug reimbursement program for MM patients (B.54) and Emergency Access to Drug Technologies (RDTL) between July 2019 and March 2024 at the Provincial Multi-Specialized Oncology and Trauma Center, Lodz, Poland. Patients treated within clinical trials were excluded from this analysis. The study adhered to the Declaration of Helsinki and received approval from the local ethical committee (The Ethical Committee of the Medical University of Lodz, No RNN/103/16/KE). Inclusion and exclusion criteria, along with drug dosing and monitoring, were established based on the reimbursement program guidelines [17]. For a brief period, Dara was administered intravenously in a dose of 16 mg per kg of body weight, up until March 2022, when the subcutaneous form of the drug was reimbursed.

According to the Polish reimbursement program, the first Dara-based regimen reimbursed in Poland was Dara, bortezomib, and dexamethasone (DVd) in RRMM in July 2019. In January 2023, Dara, bortezomib, thalidomide, and dexamethasone (DVTd) in transplant-eligible NDMM and Dara, lenalidomide, and dexamethasone (DRd) for RRMM were reimbursed. Finally, DRd in NDMM was reimbursed in Poland in January 2024. Treatment response and relapse/progression events were classified according to the International Myeloma Working Group (IMWG) criteria [18,19].

### 2.2. Antimicrobial Prophylaxis and Infectious Complications Assessment

Each patient received antimicrobial prophylaxis, which included acyclovir for herpetic infections and trimethoprim/sulfamethoxazole for Pneumocystis jiroveci infection. Intravenous immunoglobulins (IVIGs) were not routinely used for primary prophylaxis; they were mainly used for patients with life-threatening infections and those with severe, recurrent infections and low IgG levels.

Infectious events were evaluated using the Terminology Criteria for Adverse Events (CTCAE) version 5.0 [20]. It is important to note that in NDMM, ICs were assessed only during the first four cycles of DVTd before autologous hematopoietic stem cell transplantation (AHSCT) to eliminate the impact of the transplant on infection susceptibility. All patients who experienced grade 3, 4, and 5 events were hospitalized and received treatment in our Center or local hospitals if urgent hospitalization was necessary. Diarrhea was considered in this study as IC only if microbiological etiology was confirmed and/or empirical antimicrobial treatment was administered. Bacteremia was confirmed by positive blood culture, defined as the growth of microorganisms in one or more bottles, except for coagulase-negative staphylococci (CoNS), which required two positive blood cultures with matching antibiograms to confirm bacteremia. Three blood cultures were performed using media for aerobic bacteria, anaerobic bacteria, and fungi in case of fever occurrence. Sepsis was identified by an acute increase of 2 points or more in the Sequential Organ Failure Assessment (SOFA) score, along with a suspected infection. Grade 2 infectious events were identified and diagnosed through physical examinations and laboratory tests conducted during scheduled visits at the Hematology Outpatient Clinic or at the patient’s primary care physician’s office if that was their initial point of care. Testing for severe acute respiratory syndrome coronavirus 2 (SARS-CoV-2) infection was conducted using reverse transcription and real-time PCR amplification with either the Allplex SARS-CoV-2 Assay CE-IVD kit (Seegene, Seoul, Republic of Korea) or an antigen test that detects SARS-CoV-2, respiratory syncytial virus, and influenza A/B (AllTest Biotech, Hangzhou, China).

### 2.3. Statistical Analysis

The distribution of continuous variables was first verified using the Shapiro–Wilk test. The mean and standard deviation (SD) were reported for normally distributed variables, while the median and interquartile range (IQR) were used for non-normally distributed variables. The impact of clinical variables on the development of severe ICs was assessed by univariable and multivariable logistic regression models. A receiver operating characteristic (ROC) curve and area under the curve (AUC) were used to evaluate the classification effectiveness of the final model in predicting severe ICs. The optimal cutpoint on the ROC curve was selected using the Youden method. The goodness of fit for the model was evaluated using the Hosmer–Lemeshow statistic, with higher *p*-values indicating a better fit. All statistical analyses were performed using Statistica 13.3 (TIBCO, Palo Alto, CA, USA), and *p*-values less than 0.05 were considered significant.

Using our data, we developed classification models for predicting the occurrence of the most commonly reported IC of Dara-based regimens—pneumonia. The first model was trained using the J48 decision tree algorithm in the WEKA software [21]. A stochastic gradient boosting (GB) model and random forest (RF) model for pneumonia were constructed in Python 3.11.4 using the scikit-learn package version 1.5.2. The original dataset was provided in Appendix A. Due to the underrepresentation of patients with pneumonia in our set of patients (21 vs. 118 cases), the Synthetic Minority Oversampling Technique (SMOTE) was implemented to balance the dataset [22].

## 3. Results

### 3.1. Study Group Characteristics

The study group was constituted of two cohorts and included a total of 139 patients. The study group included 49 NDMM and 90 RRMM. In the NDMM group, the distribution of sex was balanced, with 24 patients (49%) being male, and 25 (51%) being female, with a median age of 63.6 years (IQR: 56.0–66.8) at the treatment commencement. Among them, forty-eight patients (98%) were treated with the DVTd regimen, and only one patient received DRD due to late reimbursement DRD in transplant-ineligible NDMM in Poland. In the RRMM group, the median age at treatment initiation was 67 years (IQR: 61.0–71.8). The majority of patients (55, 60%) received the DVd regimen, followed by DRd (31, 29.9%). The remaining demographic, clinical, and laboratory characteristics of the MM patients included in the study are presented in Table 1.

### 3.2. Profile of ICs

In the study cohort, 55 (39.6%) patients had ICs, and in total, 72 infectious events occurred. Pneumonia (24, 33.3%) was the most common, followed by upper respiratory tract infections (23, 31.9%). Other types of infections are specified in Figure 1. Most patients had grade 2 infection (21, 38.2%), followed by grade 3 (11, 20.0%), grade 1 (9, 16.4%), and grade 4 (6, 10.9%), and eight patients (14.5%) died due to infectious complications (grade 5). Overall, 25 (45.5%) patients had severe infections that required hospitalization (grade > 2). Notably, 14 (56.0%) severe infections occurred during the first months since treatment commencement, with the median time to ICs onset of 1 month (IQR: 1–2 months).

Fourteen patients required intensive treatment (grade 4 and above), mostly due to severe pneumonia (nine cases). In this group, most patients were treated empirically with IV antibiotics. In one fatal case after initial empirical therapy, antibiotics were modified to targeted IV antibiotics indicated by positive *Staphylococcus aureus* culture, two patients required targeted anti-fungal agents (one case of *Aspergillus* sp. pneumonia, one case of Candida glabrata pneumonia), and one patient received remdesivir in life-threatening SARS-CoV2 pneumonia. Sepsis occurred in five patients; non-fatal cases included one positive *Escherichia coli* culture and one coinfection caused by both *Enterococcus faecalis* and *Escherichia coli*. Three cases of fatal septic shock were reported. In one case, the microbiological investigation revealed a positive culture for *Staphylococcus haemoliticus*. Two more fatal cases of sepsis (one of probable urinary tract origin) were reported; however, no microbiological confirmation was available in this case. Two patients developed pelvic and/or abdominal abscesses and required surgical intervention, one of which died in the postoperative period due to surgical complications. The detailed clinical data on all ICs requiring hospitalization in our study cohort are provided in Table 2.

### 3.3. Factors Influencing Severe IC Occurrence

In univariable logistic regression analyses, an ECOG score above 1 (ECOG > 1) was an adverse risk factor for severe infection (OR = 5.61, 95% CI: 2.13–14.77, *p* < 0.001), whereas hemoglobin level at the treatment commencement showed a protective effect (OR 0.73, 95% CI: 0.60–0.90, *p* = 0.0036). Other laboratory variables measured just before treatment initiation, including neutrophil count, albumins, and gamma globulin levels, were insignificant in univariate analyses. In addition, clinical factors, including age, disease status (ND or RR), and gender, were also insignificant. Detailed results are provided in Table 3.

Then, we constructed a multivariable model that included significant variables in the univariable analysis adjusted by group (ND or RR) and age above 65 years old. In this model, hemoglobin level remained a significant protective factor (OR 0.77, 95% CI: 0.61–0.96, *p* = 0.0182), and ECOG > 1 was an independent predictor associated with a higher risk of infection (OR 4.46, 95% CI: 1.63–12.26, *p* = 0.0037) (Table 4). The Hosmer–Lemeshow test for goodness of fit indicated a good fit of the model (*p* = 0.6075).

This model achieved an AUC of 77% (95% CI: 66–89%) on the training dataset, with sensitivity and specificity at the optimal cutoff of 64% and 82%, respectively (Figure 2). After 10-fold cross-validation, the model preserved its discriminative potential with an AUC of 71% (95% CI: 59–83%).

### 3.4. Development of Classification Models Predicting Pneumonia Occurrence

In the next step, we aimed to develop a classification model to predict the occurrence of pneumonia, one of the most common ICs following Dara-based regimens [13]. First, we performed univariate analyses to identify potential features for model construction. Variables with a *p*-value of less than 0.2 were selected for inclusion, specifically ECOG performance status >1 (OR 4.78, 95% CI: 1.74–13.16, *p* = 0.0024), platelet distribution width (PDW) (OR 0.61, 95% CI: 0.43–0.86, *p* = 0.0048), hemoglobin level (OR 0.80, 95% CI: 0.65–0.98, *p* = 0.0331), and aspartate aminotransferase (AST) (OR 1.01, 95% CI: 1.00–1.03, *p* = 0.1936) (Appendix A). In detail, we identified a statistically significant difference in PDW between the pneumonia group (median 9.8 fL, IQR: 8.7–10.8 fL) and the non-pneumonia group (median 11.5 fL, IQR: 10.0–12.8 fL; Mann–Whitney U test, *p* = 0.0011) (Appendix A). Additionally, patients with pneumonia more frequently had PDW values below the normal range (42.1% vs. 17.4%; defined as below 9.8 fL for women and below 10 fL for men, chi-squared test with Yates correction, *p* = 0.0316).

To address the class imbalance, we used the SMOTE tool to balance the dataset. Then, the J48 algorithm was employed to train the classification model, and the final decision tree is depicted in Figure 3.

Following 10-fold cross-validation, the model achieved a sensitivity of 81.9% (95% CI: 73.19–88.74%) and a specificity of 74.6% (95% CI: 65.74–82.14%) in predicting pneumonia among MM patients treated with Dara-based regimens. The AUC in the ROC analysis was 0.77 (Figure 4). The developed model in WEKA is provided in Appendix A.

Then, we implemented other machine learning methods—GB and FR (Appendix A). All models generally had comparable sensitivity—GB: 81.9% (95% CI: 73.19% to 88.74%) and RF: 82.9% (95% CI: 74.27% to 89.51%). However, GB and RF models had higher specificity—80.5% (95% CI: 72.20% to 87.22%) and 81.4% (95% CI: 73.14% to 87.93%, respectively). In addition, the models had significantly higher AUC in ROC analyses compared to J48 (Figure 4)—GB AUC: 0.87 (95% CI: 0.82–0.91), DeLong’s test *p* = 0.0021; RF AUC: 0.87 (95% CI: 0.82–0.91), DeLong’s test *p* = 0.0009). There was no statistically significant difference between the AUC of GB and RF (*p* = 0. 9791).

## 4. Discussion

In this single-center study, we evaluated the occurrence of infectious complications (ICs) in multiple myeloma (MM) patients treated with a wide range of daratumumab-based regimens, both in newly diagnosed (ND) and relapsed/refractory (RR) populations. In our cohort, 39.6% of patients experienced at least one infectious complication during treatment, and 18% of all patients experienced severe ICs. The most common infections were pneumonia (37.5%), upper respiratory tract infections (26.8%), and urinary tract infections. Our analysis identified two key independent predictors of severe infection incidence during treatment: advanced ECOG performance status and anemia. Moreover, we were able to develop a model predicting pneumonia risk with satisfactory diagnostic parameters using four variables—ECOG performance status and PDW, hemoglobin, and AST levels.

Most patients in our cohort received the DVd regimen for RRMM and the DVTd regimen for NDMM. Overall, the frequency of infections observed in our cohort is generally comparable to those reported in pivotal randomized controlled trials [4,5,6,7,8]. For example, in the CASTOR trial, among patients treated with the DVd regimen, the incidence of any grade of pneumonia was 11.9%, and upper respiratory tract infections occurred in 24.7% of patients [5]. The rate of grade 3 or 4 infections and infestations in this trial was 21.4%. Similarly, in the CASSIOPEIA trial evaluating DVTd in NDMM, infections were reported as an adverse event of interest in 65% of patients, with grade 3 and grade 4 infections observed in 22% [8]. However, it should be noted that in NDMM patients, our study evaluated ICs only during the four-cycle pre-transplant induction period. The rationale for this was to eliminate the impact of AHSCT on the reported IC data. Additionally, DVTd was only recently reimbursed in Poland (January 2023), and the majority of patients in this subgroup are either currently undergoing or have just completed AHSCT. This may explain why the frequencies of infections reported in our study are slightly lower than those observed in the CASSIOPEIA trial. In addition, similar frequencies of infectious complications were observed in other real-life studies [23].

The key factors that increase the risk of infections in MM appear to be the stage and aggressiveness of the disease, combined with the patient’s frailty. A 2018 model developed by a Danish group on a cohort of 2557 MM patients classified individuals into high-risk (24%) and low-risk (7%) groups for severe early infections. Male sex and high tumor burden, as indicated by higher ISS stages (II and III) and elevated LDH levels, were linked to pneumonia risk, while high tumor burden and elevated serum creatinine predicted sepsis [24]. A study focused on infection risk during daratumumab treatment reported that infectious complications predominantly occurred in patients who had not achieved at least a very good partial response (VGPR) [25]. Poor performance status is a well-established factor associated with a higher incidence of overall ICs in MM patients, particularly in those treated with Dara [26,27]. Moreover, many studies have consistently shown that poor performance status, often reported as a high ECOG score, is a significant predictor of infectious complications in MM patients [28,29,30,31,32]. In our study, we observed similar rates of infections in both NDMM and RRMM patients. This aligns with existing data on novel therapies in MM, indicating that the risk of infections is more Dara-specific rather than influenced by the disease stage [33].

Our findings suggest that low hemoglobin levels in MM patients treated with Dara-based regimens are associated with an increased risk of developing infections. In MM, anemia indicates disease severity and adversely impacts quality of life, physical performance, and cardiovascular health, potentially leading to compromised immune function [34]. Anemia in MM patients is typically normocytic/normochromic, with normal to low iron levels, elevated ferritin, and significant hemosiderin deposits in bone marrow macro-phages, indicating impaired iron release from reticuloendothelial macrophages. This type of anemia is primarily driven by hepcidin, an iron-regulating hormone produced by hepatocytes, which binds to the iron exporter ferroportin, leading to its degradation and reduced iron efflux [35]. Inflammation-induced hepcidin restricts iron availability for erythropoiesis, contributing to anemia [36].

Increasing evidence suggests a link between anemia and the development of infectious diseases [37]. Reduced hemoglobin levels decrease the concentration of respiratory enzymes and mitochondrial oxidase, leading to hypoxia and lactic acid buildup. These metabolic changes impair immune response and phagocytosis, suppressing immune function and increasing infection susceptibility [38]. Lactate suppresses explicitly T-cell proliferation, cytokine release, and the cytotoxic functions of NK and CD8+ T cells [39]. In addition, iron, a critical immunomodulating nutrient, plays a vital role in regulating both humoral and cellular immune responses. Consequently, iron deficiency anemia has been associated with impaired immunogenic mechanisms, including cytokine activity, humoral and cell-mediated immunity, and non-specific immune responses [40]. Given that infections heighten the body’s iron requirements for immune functions, individuals with iron deficiency anemia may have an increased risk of infection-related mortality [41]. Emerging evidence suggests even more direct connections between anemia and immune impairment. Zhao et al. demonstrated that anemia is associated with a marked reduction in CD8+ T cell responses to pathogens in untreated mice with late-stage tumors [42]. The study highlighted the presence of immunosuppressive CD45+ erythroid progenitor cells (CD71 + TER119+), which may contribute to weakened T-cell responses commonly seen in late-stage cancer patients. These CD45+ progenitor cells, activated by tumor-induced extramedullary hematopoiesis, accumulate in the spleen and mediate immunosuppression primarily through reactive oxygen species production. The presence of a similar population of immunosuppressive CD45+ erythroid precursor cells was also confirmed in anemic cancer patients [42]. Recently, these observations have been expanded to explore how established tumors fundamentally manipulate hematopoiesis and immune responses, thereby reducing the efficacy of immune checkpoint inhibitor therapies [43]. Tumors exploit erythroid precursor cells in the spleen and bone marrow through at least two interconnected mechanisms: first, by blocking the normal differentiation pathways leading to red blood cells, and second, by dedifferentiating these lineage-committed erythroid precursors into a unique multipotent progenitor stage. These progenitor-like cells tend to transdifferentiate into the myeloid lineage [43]. This process creates a feedback loop where sustained anemia repeatedly triggers extramedullary erythropoiesis; however, the tumor-hijacked extramedullary erythropoiesis fails to replenish the necessary red blood cells. Ultimately, this results in the continuous production of erythroid-differentiated myeloid cells, leading to systemic immunosuppression and impaired immune surveillance.

Anemia has been identified as a risk factor for various infections across different groups of MM patients. In a recent study by Stevenson et al. in a large group of NDMM (N = 2030) patients, patients having hemoglobin < 10.0 g/dL were more likely to have an infection within the first 3 months of diagnosis than those having hemoglobin >10.0 g/dL (30.4% vs. 23.6%, respectively, *p* = 0.011) [44]. In this study, among anemics, hypercalcemia and renal failure were also related to susceptibility to infection. Similarly, anemia was a factor contributing to infection in several studies in NDMM [29,45]. Dumontet et al., using data from the FIRST trial, where lenalidomide and dexamethasone were compared to melphalan–prednisone–dexamethasone (MPT) in transplant-ineligible NDMM patients, included hemoglobin levels, along with ECOG performance status, lactate dehydrogenase, and serum β2-microglobulin, in a model predicting first treatment-emergent grade 3 infections within the initial four months of treatment [46]. On the other hand, a multivariate analysis of a smaller RRMM patient cohort treated with the Rd regimen identified low hemoglobin (<10 g/dL) as an independent risk factor for infections, alongside the baseline number of circulating CD3 + CD4 + CD161+ cells [47]. We have also observed the increased risk of infection in RRMM treated with lenalidomide–dexamethasone regimen [48].

To develop classifiers for pneumonia occurrence in MM patients treated with daratumumab-based therapies, we implemented three machine learning models: J48 decision tree, GB, and RF. J48 offers significant advantages in terms of interpretability. The decision trees generated by J48 provide a clear and visual representation of the decision-making process, making it easier for clinicians to understand and apply the algorithm in clinical practice. This interpretability can facilitate quick decision-making at the bedside, an essential aspect in clinical settings. This transparency not only enables them to justify their decisions to patients and their families but also enhances patient engagement and understanding. In contrast, other classifiers, such as RF and GB, often come at the cost of reduced interpretability and higher computational requirements, which may limit their practical implementation. J48 has been successfully implemented in various clinical scenarios [49,50,51,52]. However, J48 usually has lower predictive accuracy than the other proposed methods [53].

In our study, we evaluated three different classifiers, and all have almost identical sensitivity (J48: 81.9%, GB: 81.9%, and RF: 82.9%). In the context of our study, sensitivity is essential, as it helps identify multiple myeloma patients at risk of infectious complications, allowing for the timely implementation of preventive measures. Given that the primary goal of our study was early identification of at-risk individuals, the comparable sensitivity across models further supports the choice of J48. Overall, the ease of implementation of the decision tree at the bedside and its equal sensitivity were key reasons for including J48 as one of the final models for pneumonia.

Our models predicting pneumonia risk included PDW, which quantitatively assesses the heterogeneity in platelet size and is a biomarker of platelet activation [54]. Platelets have recently been increasingly recognized as important players in immune responses, capable of detecting, interacting with, and eliminating pathogens [55]. As central agents in both immune defense and coagulation, platelets are essential to the process of immunothrombosis, where immune cells initiate the formation of microthrombi to contain pathogens and facilitate their clearance from the body [56]. Platelets contain three types of granules: alpha granules (which carry chemokines, coagulation factors, PDGF, TGF-β, P-selectin, fibrinogen, and fibronectin), dense granules (containing calcium, magnesium, ADP, ATP, serotonin, and histamine), and lysosomal granules (holding enzymes like cathepsin and acid phosphatase) [57]. Alpha granule-derived P-selectin supports leukocyte adhesion to the endothelium, while dense granules release serotonin, promoting neutrophil recruitment in acute inflammation, and calcium and magnesium ions are critical in signal transduction [58]. Lysosomal enzymes contribute to immune defense by breaking down pathogens [59]. During activation, platelets undergo morphological changes to increase surface area, shifting from a discoid to a spherical shape and forming pseudopodia [60]. Larger platelets, which typically have more and larger pseudopodia, are generally more reactive than smaller ones [54]. In our study, we found that a higher PDW was associated with a reduced incidence of pneumonia in MM patients treated with Dara-based regimens. In fact, we observed that nearly half of the patients with pneumonia had PDW values below the reference range. Previous research has also noted a significantly lower median PDW (12.6 fL) in NDMM patients compared to a median PDW of 13.1 fL in healthy controls [61].

Previous studies have reported distinct patterns of PDW alteration across various inflammatory conditions. PDW has been found to be lower in several inflammatory diseases, particularly when compared to healthy controls or states of remission. These conditions include chronic spontaneous urticaria, diabetic foot ulcers, stable chronic obstructive pulmonary disease (COPD), ulcerative colitis (UC), Crohn’s disease (CD), polymyositis, and during active inflammation in rheumatoid arthritis [62,63,64,65,66,67]. Conversely, multiple studies have linked elevated PDW levels to poorer outcomes in severe infections [68,69]. In one systematic review examining the severity and mortality of COVID-19, the majority of studies found significantly increased PDW values in the non-survivor group [70]. Despite these observations, the underlying mechanisms driving the increases or decreases in PDW levels during infectious and/or inflammatory diseases remain unclear.

It is important to emphasize that our study evaluated PDW at the commencement of treatment with daratumumab, in contrast to most studies that assessed this biomarker at the diagnosis of infectious complications. Given this distinction, we believe our findings may be more indicative of potential susceptibility to infection rather than the state that occurred during the infection. We hypothesize that a higher PDW (or a PDW within the normal range) prior to treatment may reflect a preserved and effective hematopoietic response to inflammatory stimuli, whereas a reduced PDW may be associated with increased susceptibility to infection. Patients with a higher number of more reactive or immature (i.e., larger) platelets, which contribute to a higher PDW, may have a lower risk of infection during daratumumab treatment. Consistent with our observations, one study found that a lower preoperative PDW measured three days before surgery predicted postoperative sepsis in patients with colorectal cancer [71]. However, further studies are needed to confirm these hypotheses.

The infiltration of plasma cells disrupts normal hematopoiesis, often resulting in thrombocytopenia in MM patients. This condition is likely due to the physical displacement of bone marrow by plasma cells and their specific biological interactions, as suggested by the reduced platelet lifespan observed in MM [72]. It was shown that platelet reactivity is increased in MM, resulting in elevated fibrinogen binding, altered receptor expression, elevated levels of aggregation, and enhanced sensitivity to agonist stimulation [73]. Recent studies have revealed complex interactions to maintain platelet levels in MM, and compensatory cytokines like thrombopoietin (TPO), IL-6, and IL-1 are released [74]. However, these cytokines also appear to contribute significantly to the progression of MM by facilitating disease establishment through the production of adhesion molecules and promoting anti-apoptotic processes within the bone marrow microenvironment [75]. Indeed, thrombocytopenia was recently described as being associated with a more advanced stage at diagnosis and an independent factor associated with inferior survival in NDMM [76]. In accordance with our observations, low PDW was associated with poor outcomes in patients with severe pneumonia and was observed in patients with COVID-19 [77,78].

Our study has several practical implications. We identify easily recognizable risk factors for infections across a broad range of daratumumab-based regimens, encompassing both NDMM and RRMM patient populations. In particular, we demonstrate the utility of a simple decision tree with easily available features in predicting pneumonia. This information may help facilitate more personalized prophylactic strategies. Currently, there are several prophylactic options available, with levofloxacin and intravenous immunoglobulins (IVIG) being the most prominent [26]. Levofloxacin prophylaxis is particularly recommended for high-risk NDMM patients during the first three months following treatment initiation. In a randomized controlled trial, levofloxacin significantly reduced the risk of febrile episodes and death, with a hazard ratio (HR) of 0.66 (95% CI: 0.51–0.86) [79]. For patients with life-threatening or severe recurrent infections and low IgG levels, IVIG is recommended [26]. However, in our study, we could not assess the effectiveness of IVIG in reducing IC occurrence, as it was not routinely used for primary prophylaxis. Only a limited number of patients (N < 10) received IVIG, and all were administered after the occurrence of at least one infection.

Our study has several limitations. Firstly, as it is a retrospective analysis based on medical chart reviews, there may be an underestimation of lower-grade infections since not all mild ICs were reported by patients during their visits to the hematology outpatient clinic. Additionally, the small number of patients receiving prophylactic interventions, such as IVIG, limited our ability to evaluate their effectiveness.

During the development of our classifier for pneumonia occurrence, we had to address the problem of class imbalance, i.e., the difference in frequencies of patients with and without the disease. SMOTE is a well-established and frequently used method for addressing class imbalance in datasets. Without taking steps to correct the imbalance, the model would likely favor the majority class, resulting in poor sensitivity for detecting pneumonia. Initially, we empirically tested various classifiers using Cost-Sensitive Learning and adjustments to class weights. However, these approaches did not produce satisfactory results. We acknowledge that using SMOTE can generate synthetic samples that may not fully capture the variability of real cases, potentially affecting the model’s generalizability to new data, i.e., leading to overfitting [80]. Specifically, the synthetic samples produced by SMOTE could introduce noise and lead to a model that learns specific patterns of the training data rather than the underlying distribution of the actual patient population [81]. This could result in a model that performs well on training data but poorly on unseen data. In addition, it may oversample uninformative samples, and it can increase overlap between classes near their boundaries. These issues arise because SMOTE performs oversampling indiscriminately, focusing solely on the number of samples and the proximity of minority class samples while neglecting other potentially important data characteristics [82]. This all highlights the need for caution when interpreting model performance derived from SMOTE-enhanced datasets.

To mitigate this risk, we implemented several strategies. First, we preselected variables for classifier development using the original dataset before SMOTE implementation. We evaluated the performance of the final models using cross-validation, which helps reduce bias by ensuring that each fold includes both original and synthetic samples, thereby promoting a more robust assessment of the model’s performance. Additionally, we reported multiple performance metrics, including sensitivity, specificity, and the area under the receiver operating characteristic (ROC) curve, to provide a comprehensive evaluation beyond simple accuracy. While the ideal approach would be to perform external validation of our model on a new dataset, to the best of our knowledge, no such dataset is currently available. We acknowledge this as a limitation and suggest that future studies should aim to validate the findings using independent clinical datasets if they become available.

Despite these constraints, our study offers a cohort of patients treated with a broad range of Dara-based regimens, highlighting easily identifiable variables that consistently indicate the risk of ICs.

## 5. Conclusions

In conclusion, our study demonstrates the importance of simple, effective clinical and laboratory assessments, such as hemoglobin levels and ECOG performance status, in identifying patients susceptible to infections during treatment with Dara-based regimens. By incorporating these evaluations into routine clinical practice, healthcare providers can more accurately identify vulnerable patients and implement personalized prophylactic strategies to mitigate infection risk. This approach not only enhances patient care but also has the potential to improve overall treatment outcomes in MM patients undergoing daratumumab therapy.

## Figures and Tables

**Figure 1 cancers-16-03709-f001:**
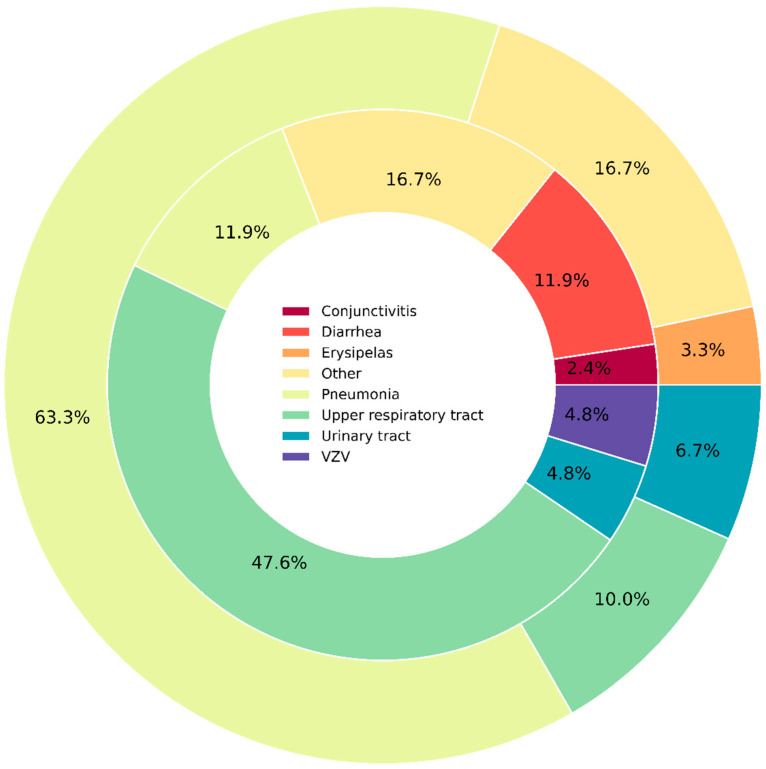
Profile of infectious complications (ICs) in a study cohort. The inner cycle represents mild infections, whereas the outer represent severe infections (grade > 2). VZV—Varicella zoster virus.

**Figure 2 cancers-16-03709-f002:**
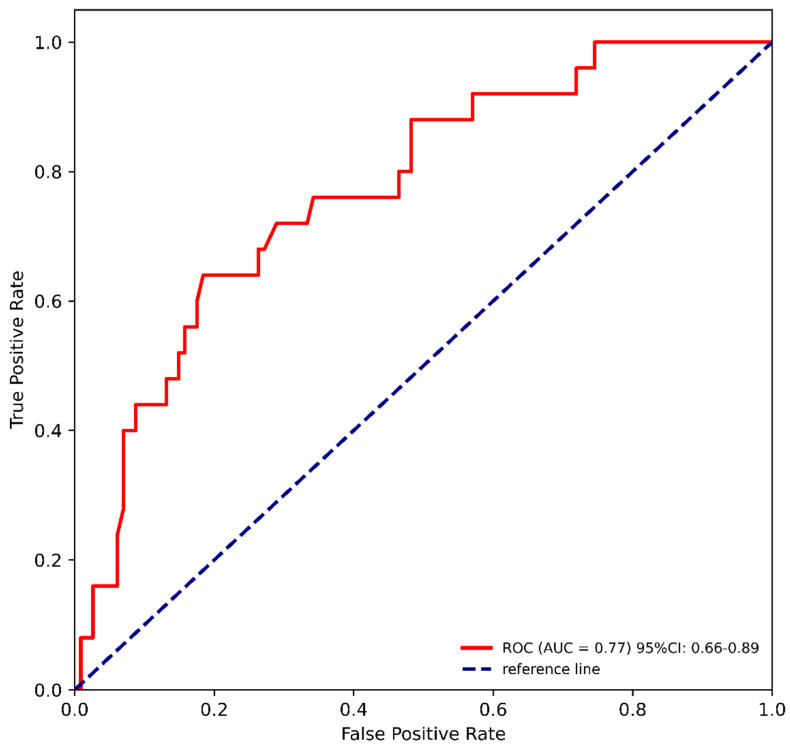
The ROC curve of the multivariable logistic regression model on the training dataset discriminating high-risk multiple myeloma (MM) patients for infectious complications (ICs) treated with daratumumab.

**Figure 3 cancers-16-03709-f003:**
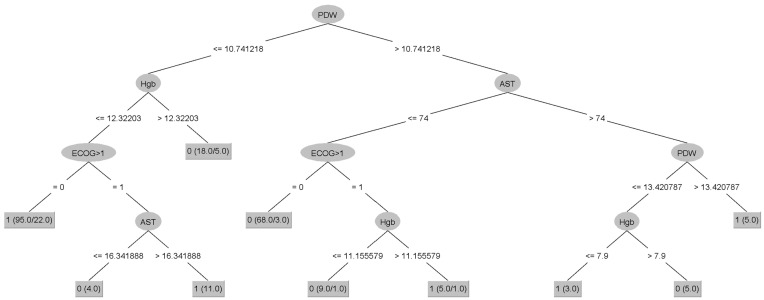
J48 decision tree with model separating multiple myeloma (MM) patients who experienced pneumonia during treatment with a daratumumab-based regimen. AST—aspartate aminotransferase; ECOG—Eastern Cooperative Oncology Group; Hgb—hemoglobin; PDW—platelet distribution width.

**Figure 4 cancers-16-03709-f004:**
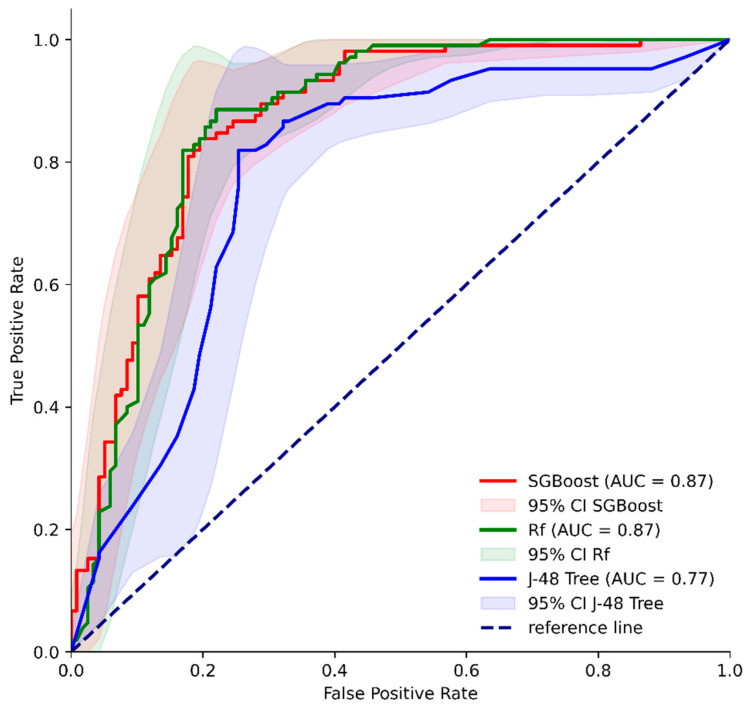
ROC curves and AUC comparison for developed classifiers for pneumonia occurrence in multiple myeloma (MM) patients treated with daratumumab-based regimens.

**Table 1 cancers-16-03709-t001:** The characteristics of the study group, comprising two cohorts of patients treated with daratumumab-based regimens—newly diagnosed multiple myeloma (NDMM) and relapsed/refractory multiple myeloma (RRMM). Unless otherwise specified, all data are presented as frequency and percentage (%).

Characteristics	NDMM	RRMM
Number of patients	49 (100)	90 (100)
Gender		
Female	25 (51.0)	36 (40.0)
Male	24 (49.0)	54 (60.0)
Age at diagnosis median (IQR)	63.6 (56.0–66.8)	61.5 (56.2–67.0)
Age at administration median (IQR)	63.6 (56.0–66.8)	67.0 (61.0–71.8)
Previous treatment lines median (IQR)	-	2 (2–3)
R-ISS/ISS *		
1	7 (14.3)	20 (22.2)
2	24 (49.0)	25 (27.8)
3	11 (22.4)	31 (34.4)
Missing	7 (14.3)	14 (15.6)
Cytogenetic risk (IMWG)		
High risk	12 (24.5)	9 (10.0)
Standard risk	31 (63.3)	16 (17.8)
Missing	6 (12.2)	65 (72.2)
Previous AHSCT		
No	-	34 (37.8)
Yes	-	56 (62.2)
Paraprotein		
IgG kappa	13 (26.5)	31 (34.4)
IgG lambda	10 (20.4)	18 (20.0)
IgA kappa	9 (18.4)	14 (15.6)
LCD lambda	4 (8.2)	10 (11.1)
LCD kappa	4 (8.2)	9 (10.0)
Ig A lambda	4 (8.2)	4 (4.4)
Biclonal	3 (6.1)	2 (2.2)
Non-secretory	1 (2.0)	1 (1.1)
Missing	1 (2.0)	1 (1.1)
Treatment regimen		
DRD	1 (2.0)	31 (34.4)
DVD	0 (0.0)	55 (61.1)
DVTd	48 (98.0)	0 (0.0)
Monotherapy	0 (0.0)	4 (4.4)
ECOG		
0	1 (2.0)	2 (2.2)
1	36 (73.5)	74 (82.2)
≥2	11 (22.4)	14 (15.6)
Missing	1 (2.0)	0 (0.0)

*—For NDMM, the R-ISS is provided, while for RRMM, the ISS is used due to the absence of cytogenetic data. AHSCT—autologous hematopoietic stem cell transplantation; DRD—daratumumab, lenalidomide, and dexamethasone; DVD—daratumumab, bortezomib, and dexamethasone; DVTd—daratumumab, bortezomib, thalidomide, and dexamethasone; ECOG—Eastern Cooperative Oncology Group; IMWG—International Myeloma Working Group; ISS—International Staging System; LCD—light chain disease; R-ISS—Revised International Staging System.

**Table 2 cancers-16-03709-t002:** The clinical data of severe infectious events in the study cohort (25 cases).

Group	Age	Gender	CTCAE Term	No of Cycle	Etiology	Treatment	Grade Acc. to CTCAE v5.0
RR	67	Male	Pneumonia, septic shock	1	*Aspergillus* sp., unknown	Antibiotic and antifungal IV	5
ND	55	Male	Perforated pelvic abscess, peritonitis	2	Unknown	Surgical	5
ND	60	Female	Sepsis	1	*Staphylococcus* *epidermidis*	Antibiotic IV	5
RR	73	Male	Pneumonia	1	*Candida glabrata*	Antibiotic and antifungal IV	5
RR	56	Male	Pneumonia	2	Unknown	Antibiotic IV	5
ND	53	Female	Pneumonia	2	*Staphylococcus aureus*	Antibiotic IV	5
RR	57	Male	Pneumonia followed by sepsis	1	*Staphylococcus haemolyticus*	Antibiotic IV	5
RR	70	Male	UTI, sepsis	1	Unknown	Antibiotic IV	5
ND	67	Female	Pelvic abscess	3	Unknown	Surgical	4
RR	72	Female	Pneumonia	1	Unknown	Antibiotic IV	4
RR	80	Female	Pneumonia	1	Unknown	Antibiotic IV	4
ND	65	Male	Pneumonia	1	SARS-CoV2	Remdesivir	4
ND	65	Female	Pneumonia followed by sepsis	2	*Escherichia coli*	Antibiotic IV	4
ND	51	Female	Sepsis	1	*Enterococcus* *faecalis* *Escherichia coli*	Antibiotic IV	4
RR	60	Female	Pneumonia	1	Unknown	Antibiotic IV	3
RR	66	Male	Erysipelas	2	*Streptococcus* *pyogenes*	Antibiotic IV	3
RR	67	Female	Pneumonia	2	Unknown	Unknown	3
RR	64	Male	Pneumonia	4	Unknown	Unknown	3
RR	69	Male	Pneumonia	4	Unknown	Unknown	3
RR	67	Male	Pneumonia	1	Unknown	Antibiotic IV	3
ND	69	Male	Pneumonia	2	Unknown	Unknown	3
ND	69	Female	Pneumonia	1	Unknown	Antibiotic IV	3
ND	66	Male	Pneumonia	1	Unknown	Antibiotic IV	3
ND	57	Male	Pneumonia	1	SARS-CoV2	Symptomatic treatment	3
RR	69	Male	UTI	1	*Enterococcus* *faecalis* *Escherichia coli*	Antibiotic IV	3

IV—intravenous; RR—relapsed/refractory; ND—newly diagnosed; UTI—urinary tract infection.

**Table 3 cancers-16-03709-t003:** Univariable logistic regression analysis of clinical and laboratory variables for severe infectious complication (IC) occurrence in multiple myeloma (MM) patients. All laboratory variables were measured before the commencement of daratumumab-based therapy.

Variable	OR	95% CI		*p*-Value	FDR
		Lower	Upper		
Age > 65	0.92	0.75	1.13	0.4234	0.6748
Gender—Male	0.55	0.09	3.55	0.5342	0.6748
Relapsed Disease	1.01	0.96	1.06	0.6979	0.7976
ECOG > 1	5.61	2.13	14.77	0.0005	0.0120
Stage 3 ISS/R-ISS *	2.27	0.93	5.53	0.0715	0.4228
Hemoglobin (g/dL)	0.73	0.60	0.90	0.0036	0.0432
RDW-SD (fL)	1.02	0.98	1.06	0.2640	0.5760
Platelets (10^9^/L)	0.99	0.99	1.00	0.0739	0.4228
MPV (fL)	0.87	0.56	1.34	0.5201	0.6748
PDW (fL)	0.80	0.62	1.04	0.0941	0.4228
WBC (10^9^/L)	1.00	0.98	1.02	0.8182	0.8572
Neutrophil Count (10^9^/L)	1.19	0.96	1.48	0.1057	0.4228
Lymphocyte Count (10^9^/L)	1.40	0.58	3.37	0.4555	0.6748
Calcium (mmol/L)	0.86	0.53	1.38	0.5310	0.6748
Creatinine (mg/dL)	0.95	0.52	1.72	0.8572	0.8572
Urea (mmol/L)	1.01	0.99	1.03	0.2449	0.5760
Albumin (g/L)	0.96	0.89	1.02	0.1993	0.5315
Gamma Globulins (g/L)	0.86	0.7	1.05	0.1389	0.4551
M Protein (g/L)	1.01	0.99	1.03	0.2941	0.5882
Total Protein (g/dL)	1.21	0.5	2.93	0.6659	0.7976
Bilirubin (mg/dL)	1.03	0.97	1.08	0.3361	0.6175
ALT (U/L)	1.01	1.0	1.02	0.1517	0.4551
AST (U/L)	0.48	0.1	2.32	0.3602	0.6175
CRP (mg/L)	0.98	0.83	1.16	0.8497	0.8572

*—For NDMM, the R-ISS is provided, while for RRMM, the ISS is used due to the absence of cytogenetic data. ALT—alanine aminotransferase; AST—aspartate aminotransferase; CRP—C-reactive protein; ECOG—Eastern Cooperative Oncology Group; FDR—false discovery rate; ISS—International Staging System; MPV—mean platelet volume; OR—odds ratio; PDW—platelet distribution width; RDW-SD—red cell distribution width–standard deviation; R-ISS—Revised International Staging System; WBC—white blood cell count.

**Table 4 cancers-16-03709-t004:** Final multivariable logistic regression model for severe infectious complication (IC) occurrence in multiple myeloma (MM) patients treated with daratumumab-based regimens.

Variable	OR	CI 95%	CI 95%	*p*-Value
ECOG > 1	4.46	1.63	12.26	0.0037
Hemoglobin (g/dL)	0.77	0.61	0.96	0.0182
Age > 65	1.57	0.65	4.48	0.3617
Relapsed Disease	0.72	0.27	1.92	0.5162

## Data Availability

The clinical data required to generate the decision tree model are provided in Appendix A. The generated classification model is included in Appendix A. More detailed source data can be made available by the corresponding author upon reasonable request.

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
