# Peer review of "Predictors and Profile of Severe Infectious Complications in Multiple Myeloma Patients Treated with Daratumumab-Based Regimens: A Machine Learning Model for Pneumonia Risk"

_cancers, 2024, doi:10.3390/cancers16213709_

Round 1
Reviewer 1 Report
Comments and Suggestions for Authors
The manuscript, entitled "Predictors and Profile of Severe Infectious Complications in Multiple Myeloma Patients Treated with Daratumumab-Based Regimens," The article, entitled "A Machine Learning Model for Pneumonia Risk," focuses on the implementation of machine learning in the predictive power of daratumumab and concomitant therapy in multiple myeloma. The study of Darа monoclonal antibody-based drug Darа is of particular significance to this work. Up until this point in time, the treatment of multiple myeloma has not yielded the desired results. Daratumumab (Dara) is the inaugural monoclonal antibody to be utilized in the treatment of multiple myeloma. Artificial intelligence is now playing an increasingly significant role in the field of medicine, with the potential to elevate diagnostic capabilities to a previously unimagined level. The work is of significant relevance and I recommend its publication in the submitted form.
Author Response
Response: We are grateful to the reviewer for the positive assessment of our work.
Reviewer 2 Report
Comments and Suggestions for Authors
The article entitled “Predictors and Profile of Severe Infectious Complications in
Multiple Myeloma Patients Treated with Daratumumab-Based Regimens: A Machine Learning Model for Pneumonia Risk” talks about the infections in 139 newly diagnosed (ND n. 49) and relapsed/refractory (RR n. 90) patients with multiple myeloma (MM). This topic is clinically important. Infact, the infections are a significant complication responsible for morbidity and mortality in MM patients. The authors conducted a exhaustive retrospetive research study and used appropriate methods. The results are intersting because the parameter of hemoglobin and the evaluation of ECOG are non-invasive measurements and accessible to all healthcare professionals. The discussioni is well-written with biological and clinical details. I only suggest adding two articles in the discussion section (Johnsurud A et al, Blood 2017; Johnsurud A et al, BJH 2019).
Author Response
Response: We thank the Reviewer for assessing our effort favorably. In accordance with the reviewer's recommendation, we broadened the discussion to incorporate a recommended reference relevant to the subject of our manuscript.
Reviewer 3 Report
Comments and Suggestions for Authors
The paper attempts to address the important issue of infectious complications in multiple myeloma patients treated with daratumumab-based regimens. However, despite the relevance of the topic, several critical issues limit its scientific contribution. The study's retrospective nature is a fundamental weakness, and the decision tree model's utility is undermined by limited external validation. Furthermore, the dataset's small size raises concerns about the reliability of the conclusions drawn from it, especially given the oversampling technique applied to handle imbalanced data. In addition, the discussion lacks depth in exploring potential biological mechanisms underlying the increased infection risk, which weakens the overall impact of the findings. The paper also suffers from an unclear presentation of statistical methods, making replication of the study difficult. These significant concerns suggest the need for a major revision before it can be considered a valuable addition to the field.
1. The use of the J48 decision tree algorithm (page 8) is inadequately justified. Although common in machine learning, this method is not well-suited for clinical data with high variability, as seen in this study. A more robust method, such as random forests or gradient boosting, could improve predictive accuracy.
2. The claim "Our decision tree model predicting pneumonia risk included PDW" (page 10) lacks clarity regarding how PDW biologically relates to infection risk. The biological rationale for including this parameter is weak, and there is no supporting literature provided.
3. The use of SMOTE to balance the dataset (page 9) could introduce bias, particularly given the limited number of cases of pneumonia (21 cases out of 139). This technique may artificially inflate the performance of the model, undermining the reliability of the results.
4. The sentence "Surprisingly, other laboratory variables...were insignificant" (page 8) implies the authors were not expecting this outcome. This suggests a lack of prior hypothesis formulation, which diminishes the scientific rigor of the study.
5. The study fails to account for potential confounding factors in the multivariable analysis, particularly with regard to the patients' prior treatments and comorbidities. This significantly limits the generalizability of the results to broader patient populations.
6. The paper lacks a sufficient exploration of the implications of low hemoglobin on immune function (page 12). Although anemia is mentioned as a risk factor, the paper misses an opportunity to delve into the pathophysiological mechanisms, which are critical to understanding infection susceptibility in multiple myeloma.
7. The description of the statistical methods (page 9) is overly simplistic. For instance, no details are provided regarding the goodness-of-fit tests or the model calibration process. Without this, it is difficult to assess the robustness of the logistic regression model.
8. The reference to "p=0.0037" for multiple variables in the final model (page 9) raises concerns about multiple testing without proper corrections. The study should use methods such as Bonferroni correction or false discovery rate to adjust for multiple comparisons.
Comments on the Quality of English Language
Moderate editing of English language required.
Round 2
Reviewer 3 Report
Comments and Suggestions for Authors
The response letter adequately addresses several reviewer concerns, but there remain areas for improvement in the explanations and methodology. The statistical methods are enhanced, and the decision tree model justification is clarified, yet the biological rationale for some of the variables remains underdeveloped. Additionally, while the inclusion of alternative classifiers strengthens the paper, the handling of imbalanced data could benefit from further elaboration.
1. The justification for using PDW as a predictor of pneumonia risk remains weak. A clearer biological rationale, beyond statistical significance, is needed to explain why this parameter is included in the model.
2. The explanation for the use of SMOTE to address class imbalance is still insufficient. While its advantages are mentioned, the potential downsides, such as overfitting and how these were mitigated, are not discussed in enough detail.
3. The comparison between the J48 decision tree and more robust models like gradient boosting (GB) and random forests (RF) could be expanded. Specifically, the trade-off between interpretability and performance deserves a more nuanced discussion, as RF and GB consistently showed higher accuracy.
Comments on the Quality of English Language
Moderate editing of English language required.
